# Timing of first prenatal ultrasound and associated factors among women who gave birth at health institutions in Ambo Town, central Ethiopia

**Samson Mesfin[1], Gizachew Abdissa Bulto[2]\*, Motuma Gutu[3], Natnael Dechasa Gemeda[1]**

1 Department of Midwifery, College of Medicine and Health Sciences, Dilla University, Dilla, Ethiopia, 2 Department of Midwifery, College of Health Sciences and Referral Hospital, Ambo University, Ambo, Ethiopia, 3 Department of Obstetrics and Gynecology, College of Health Sciences and Referral Hospital, Ambo University, Ambo, Ethiopia

\* gizachab@yahoo.com

## Abstract

### Background

The Ethiopian Ministry of Health recommends "one prenatal ultrasound scan before 24 weeks of gestation for every pregnant woman." Despite clear suggestions for timely prenatal ultrasound utilization, little is known about the extent to which it is utilized and the factors affecting the timing of the first prenatal ultrasound examination in the study area. Hence, this study aimed to assess the timing of the first prenatal ultrasound and identify associated factors among women who delivered at health institutions in Ambo town, central Ethiopia.

### Methods

This health facility-based cross-sectional study was conducted from September 12 to October 30, 2022. Data were collected through interviews using structured questionnaires and record reviews. A total of 442 participants were recruited through systematic random sampling. Data analysis was performed using a binary logistic regression model in SPSS Version 25. Adjusted odds ratios with a p-value of less than or equal to 0.05 were used to declare statistical significance.

### Results

Overall, 71% (95%CI: 67.0–75.6) of participants had received a timely prenatal ultrasound. Living in urban areas (AOR = 5.64,95%CI:2.53–12.55), having a history of prenatal ultrasound during previous pregnancy (AOR = 2.47,95%CI:1.24–4.89), attending ANC visits at hospital (AOR = 3.30,95%CI:1.19–9.16), and good knowledge of prenatal ultrasound (AOR = 4.46,95%CI:2.26–8.81) were found to significantly affecting the timing of the first prenatal ultrasound.

**Data availability statement:** All relevant data are within the paper and its Supporting information files.

**Funding:** The funding for this study was provided by Ambo University, but the funder had no role in the conceptualization, design, data collection, analysis, decision to publish, or preparation of the manuscript.

**Competing interests:** The authors have declared that no competing interests exist.

**Abbreviations:** ANC, antenatal care; GA, gestational age.

## Conclusion

In this study, more than a quarter of the women did not receive timely prenatal ultrasounds. Urban residence, previous use of prenatal ultrasound, attending ANC at the hospital, and having good knowledge were factors identified for timely prenatal ultrasound. Therefore, all stakeholders must work on those identified factors to improve the timely ultrasound scanning.

## Introduction

Timely prenatal ultrasound was one that is performed before 24 weeks of gestation. Prenatal ultrasound is an essential core package for routine Antenatal Care (ANC) [1,2]. According to the World Health Organization (WHO), every pregnant mother should undergo at least one ultrasound examination before reaching 24 weeks of Gestational Age (GA) [1]. Similarly, the International Federation of Gynecology and Obstetrics advocates for two ultrasound screenings for all pregnant women during their first and second trimesters [3]. Furthermore, the Society of Obstetricians and Gynecologists of Canada recommends offering all pregnant women an ultrasound scan between weeks 18 and 22 to detect potential fetal anomalies [4]. Prenatal ultrasound helps in the early diagnosis and management of obstetric complications such as incomplete miscarriage, ectopic pregnancy, insufficient cervix, blighted ovum, and intrauterine growth restriction. It also contributes to the identification of aneuploidy and diagnosis of structural anomalies [5–8].

Timely prenatal ultrasound scans have become a central tool in Antenatal Care (ANC) services in Ethiopia. The Ministry of Health recommends that all pregnant women use ultrasound scans timely (before 24 weeks of gestation) as routine prenatal care. It is important to diagnose multiple gestations, determine placentation, ascertain gestational age, and identify fetal anomalies. The guidelines also clarify that timely prenatal ultrasound has more benefits for early screening of Gestational Trophoblastic Disease and for accurate diagnosis of gestational age before 24 weeks of gestation (preferably crown to rump length measurement before 14 weeks) [5–11]. These conditions could potentially be addressed if all women receive timely prenatal ultrasound. Nevertheless, pregnant women in Sub-Saharan Africa have yet to fully adopt this practice. Most women in the region still experience pregnancy without the advantage of a one-time ultrasound examination. This is due to a range of factors, ranging from personal challenges in health-seeking behaviors to limitations in health-care facility capacities and lack of clear national policies [12–18].

The introduction of a timely prenatal ultrasound service at public hospitals in Ethiopia increases antenatal care visits, encourages mothers to have an institutional delivery, improves postnatal care utilization [13], and has a beneficial psychological effect on initiating bonding, reducing anxiety, and depression in the mother during pregnancy [11].

Delaying the initial prenatal ultrasound until ≥24 weeks of gestation can have various consequences. Primarily, as pregnancy advances, the accuracy of fetal

gestational age estimation becomes less reliable, with the margin of error potentially exceeding 20 days during the third trimester [9,10]. Moreover, imprecise gestational age determination may contribute to unfavorable outcomes, including low birth weight, infant and neonatal mortality [19], and undetected congenital abnormalities. This unreliable estimation of gestational age can pose challenges when managing conditions, such as preterm labor, intrauterine growth restriction, and induction of labor for post-term pregnancy [10]. Studies have also reported that the use of early ultrasound scans to determine gestational age lowers the rate of induction for postdate pregnancy [20,21].

In Ethiopia, the primary objective of the Health Sector Transformation Plan (HSTP) is to lower maternal mortality from 401 to 279 per 100,000 live births and to decrease neonatal mortality from 33 to 21 per 1,000 live births [22]. The risk of maternal and infant mortality can be mitigated by enhancing access to timely prenatal ultrasound, which is regarded as a vital component of antenatal care [1]. While national guidelines recommend early prenatal ultrasound (prior to 24 weeks of gestation), there is a dearth of information regarding its implementation in Ethiopian public health facilities. The extent of its use and factors that affect the use of timely prenatal ultrasound remain scarce. Previous studies only assessed the proportion of pregnant women who underwent ultrasound scans, not the timing of their first perinatal ultrasound. Furthermore, no data are available on this topic in the study area to guide local interventions. Consequently, the researchers conducted this study to determine the timing of initial prenatal ultrasound scans and the associated factors among women who gave birth in public health institutions in Ambo Town, Ethiopia, in 2022.

## Methods and materials

### Study design, area, and period

An institution-based cross-sectional study was conducted to assess the timing of the first prenatal ultrasound and its associated factors among mothers who delivered at public health facilities in Ambo town from September 12 to October 30, 2022. Situated 114 km from Addis Ababa, the nation's capital, Ambo Town, serves as the administrative center of the West Shewa zone. According to the Ambo Town Health Office, there are two hospitals (Ambo University Referral Hospital and Ambo General Hospital), two health centers (Ambo Health Centre, Awaro Health Centre), six health posts, and twenty-six medium and ten small private clinics that provide preventive and curative services to the community. Only Amo University Referral Hospital, Ambo General Hospital, Ambo, and Awaro Health centers provided delivery services in the town and were selected for the current study. The selected health facilities offer Antenatal Care, Delivery, and Postnatal services for 7,328 pregnant women and attend to 5,772 births per year [23].

### Populations

All randomly selected women who delivered at public health facilities in Ambo Town and had at least one prenatal ultrasound scan were the study population. Women who did not undergo a prenatal ultrasound scan, could not recall, or had no record of their first ultrasound scan during their recent pregnancy were excluded.

### Sample size and sampling procedure

The sample size for the current study was determined using Epi Info statistical software version 7.1, employing the double population proportion formula. A 5% level of significance, 80% power, and 1:1 ratio of unexposed to exposed were considered. Using the variable "history of abortion," with 71.93% of outcomes in the exposed group and 58.13% in the unexposed group, along with an AOR of 1.845 [17], the sample size was calculated to be 402. After accounting for a 10% non-response rate, the final sample size was adjusted to 442.

A systematic random sampling method was employed to recruit the respondents. The performance report from July 23 to September 5, 2022, served as a reference for proportionally allocating study subjects to health institutions. A total of 758 postpartum women were identified at public health institutions in Ambo town. Specifically, 305 women were from

Ambo University Referral Hospital, 341 from Ambo General Hospital, 45 from Ambo Health Centre, and 67 from Awaro Health Centre. Consequently, the sample was proportionally distributed across these selected health institutions: 178 (40.3%) participants were allocated to the Ambo University Referral Hospital, 199 (45.0%) to the Ambo General Hospital, 26 (5.9%) to the Ambo Health Center, and 39 (8.8%) to the Awaro Health Center. Finally, using the delivery registration book as a sampling frame, every second woman who gave birth was selected for the study, with the first participant chosen randomly (Fig 1).

### Variables of the study

In this study, the timing of the first prenatal ultrasound was used as the dependent variable, whereas a variety of independent variables were examined and categorized into several groups. Sociodemographic variables included the mother's age, place of residence, marital status, religion, ethnicity, educational attainment of both the mother and her husband, maternal occupation, and household income. The maternal factors considered were prior knowledge of timely prenatal ultrasound scans, previous experience with such scans, any health issues before or during pregnancy, whether the pregnancy was intended, alcohol consumption during pregnancy, and the presence of multiple pregnancies. Antenatal care (ANC) history variables encompassed the timing of the first ANC visit, total number of ANC visits, and partner support during pregnancy. Reproductive history variables included gravidity, parity, abortion, congenital anomalies, and premature birth. Birth-related factors included the mode and place of delivery, history of Caesarean section, and gestational age at birth. Additionally, the study evaluated the participants' knowledge and attitude towards prenatal ultrasound.

### Operational definition

The timing of the first prenatal ultrasound is considered early when the participants used an ultrasound for the first time before 24th weeks of gestational age, and late prenatal ultrasound when the participants used an ultrasound for the first time at 24th and above weeks of gestational age [1,2].

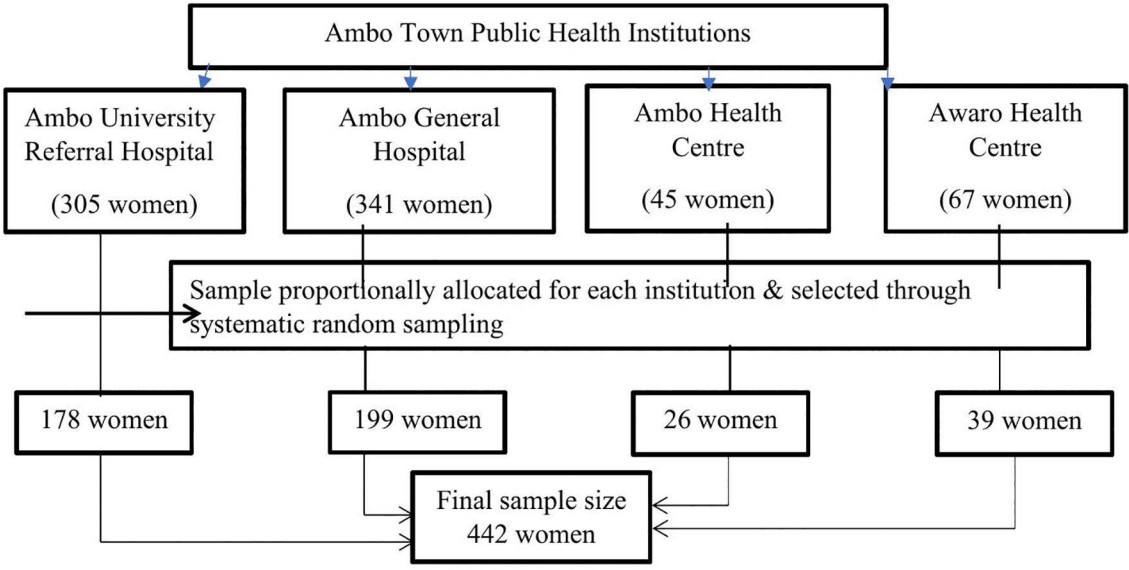

**Fig 1. Schematic presentation of the sampling procedure for the timing of first prenatal ultrasound and associated factors among women who gave birth in Ambo Town Public Health Institution, Ethiopia, 2022.**

Knowledge of prenatal ultrasound: Twelve questions with yes, no and I don't responses were used to assess participants' knowledge of prenatal ultrasound. The average score was used to determine the status with the minimum and maximum scores of 0–12. Overall knowledge status was classified as 'good' if a participant's score was equal to or exceeded the mean and 'poor' if it fell below the mean score of these knowledge-assessing items [15].

The attitude score was determined based on a composite value derived from the responses to a 5-point Likert-type scale ranging from strongly agree to strongly disagree with the score ranging from 0 to maximum of 50. The overall attitude was determined by computing the average of all ten attitude-related questions. Respondents whose scores were equal to or exceeded the mean value were classified as having a favorable attitude, while those scoring below the mean were deemed to have an unfavorable attitude towards prenatal ultrasound [15].

## Data collection tools and techniques

The survey instrument was developed through a review of relevant literature, with adaptations and contextualization to suit the local setting [14–18,24,25]. The research team, which consisted of senior Clinical Midwives and Obstetrician and Gynecologist had reviewed and approved the contents of the data-collection tool. The questionnaire was initially prepared in English and subsequently translated into the local languages of Afan Oromo and Amharic, and then back translated to English to ensure consistency (S1 File). Internal reliability of the items was evaluated using Cronbach's alpha. For the knowledge assessment, the Cronbach's alpha value was calculated at 0.87, while for the attitude assessment, it was 0.85.

Data were collected through interviews with women who had given birth in the postnatal room and by reviewing participants' medical records to minimize recall bias. The timing of the first ultrasound scan was determined using both the printed scan results and those recorded by the healthcare providers in the participants' medical records. The data were collected for a period of one month and three weeks by four data collectors with BSc in the midwifery profession who were fluent in local languages and experienced in data collection. A supervisor with a master's degree in public health supervised the data collection process.

## Data quality control for the study

A two-days training session was conducted for the data collectors and supervisors, focusing on the study's objectives, methodologies, and instruments prior to commencing data collection. A pretest was conducted to check the clarity, understandability, and simplicity of the questionnaire on 5% of mothers who gave birth at Ambo General Hospital. Following the pre-test, the questionnaire was modified based on the feedback and suggestions received. Investigators and supervisors performed daily checks on the collected data to ensure completeness and consistency. To maintain the security and confidentiality of the collected data, all questionnaires were securely stored in a locked cabinet.

## Data processing and analysis

Prior to analysis, the data were cleaned and cross-verified. Subsequently, the data were coded and entered into Epi Data software Version 3.1 and subsequently exported to SPSS Version 25.0. Descriptive analyses, including frequency distribution, measures of central tendency, and variability (mean and standard deviation), were computed to characterize the study participants in relation to relevant variables. The normality of the distribution of participants' knowledge and attitudes towards perinatal ultrasound was assessed through visual inspection of histograms and Q–Q plots. These findings demonstrated that the data were normally distributed. Binary logistic regression analysis was used to assess the presence and degree of association between the dependent and independent variables. Variables with a p-value < 0.25 at 95% confidence interval in the bivariable logistic regression were incorporated into a multivariable logistic regression. Multiple logistic regressions were conducted using the backward-LR method to identify factors associated with the outcome variable (timing of first the prenatal ultrasound).

The model goodness-of-fit was assessed using the Hosmer–Lemeshow test, and the value of the model fitness test was 0.44. Multi-collinearity was evaluated using the variance inflation factor (1.0–2.2) and tolerance (0.4–0.9). Subsequently, all candidate variables were fitted into the multivariable model, as no collinearity was observed among them. The adjusted odds ratio (AOR) with a 95% confidence interval (CI) and p-value <0.05 was considered statistically significant. Data are presented in the form of tables, graphs, and text.

## Ethics statement

Ethical clearance for the study was obtained from the Ethical Review Board of the College of Medicine and Health Sciences, Ambo University (Ref. no AU/CMHS-RCS/19/2022). A letter of support was also obtained from the Ambo Town Health Office for each health institution. Permission was obtained from the management of each health institution. The study participants were briefed on the research objectives and informed of their right to voluntarily participate or withdraw at any point without justification. All participants provided written informed consent for participation in the study. Data collection was conducted anonymously, omitting personal identifiers, and the collected data were stored securely in a locked cabinet. They were assured that the information they provided would be utilized solely for research purposes.

## Results

### Socio-demographic characteristics of respondents

A total of 442 postpartum mothers participated in the study, resulting in a response rate of 100%. The mean age of the study participant was 25.6 years with a standard deviation of ± 4.5, ranging from 18 to 42 years of age. Majority respondents 437(98.9%) were married and 209(47.3%) of were housewives. Regarding educational status, 150(33.9%) mothers and 212(48.0%) husbands' attended college or above. Additionally, 352 (79.6%) of mothers resided in urban areas, and 247 (55.9%) reported an average monthly income ranging from 1000 to 5000 ETB (Table 1).

### Timing of first prenatal ultrasound scan and related factors

The proportion of women who had a timely prenatal ultrasound were 71% [(95% CI); (67.0%−75.6%)]. The time of the first prenatal ultrasound among respondents ranged from 4.7 weeks to 42 weeks of gestation, with a mean timing of 20.7 weeks with a standard deviation of 6.38 weeks. Nearly half of the participants, 227 (51.4%), were informed about the ideal time for a prenatal ultrasound scan, and 167 (37.8%) of them received this information from a health professional (Table 2). The figure shows the proportion of women who underwent their first prenatal ultrasound scan along with their corresponding gestational age across the eight contact ANC (Fig 2).

### Obstetric and reproductive health related factors

All the respondents had received ANC services during recent pregnancy, but 27.1% began their ANC follow-up in the first trimester of pregnancy. Moreover, only 66(14.9%) mothers had five or more ANC contacts, and 138(31.2%) mothers visited ANC alone without their partners. Regarding medical history, 28(6.3%) and 51(11.5%) participants had health problems before and during pregnancy, respectively. In total, 262 respondents (59.3%) were multigravidas. Adverse obstetric history was reported by 39 respondents (8.9%), and 30 (6.8%) had a history of abortion. Furthermore, 382 mothers (86.4%) delivered in hospitals, of whom 120 (27.1%) underwent a caesarean section. Most of the babies, 423 (95.7%), were born at term, with 404 (91.4%) weighing–4, 000 g, and 46 (10.4%) were admitted to the Neonatal Intensive Care Unit (Table 3).

### Knowledge of postpartum women on prenatal ultrasound

In this study, the mean score of mothers' knowledge on prenatal ultrasound was 8.84(SD±3.28) with minimum and maximum scores of 0 and 12, respectively. This study showed that 389(68.5%) participants had a good knowledge of prenatal

**Table 1. Socio-Demographic Characteristics of Women who gave birth in Ambo Town Public Health Institution, Ethiopia, 2022. (N = 442).**

| Variable | Category | Frequency | Percentage (%) |
|---|---|---|---|
| **Age** | 18–20 | 27 | 6.1 |
| | 20–24 | 167 | 37.8 |
| | 25–29 | 154 | 34.8 |
| | 30–34 | 70 | 15.8 |
| | ≥35 | 24 | 5.4 |
| **Residency** | Urban | 352 | 79.6 |
| | Rural | 90 | 20.4 |
| **Marital status** | Married | 437 | 98.9 |
| | Other* | 5 | 1.1 |
| **Religion** | Orthodox | 134 | 30.3 |
| | Protestant | 286 | 64.7 |
| | Other** | 22 | 5.0 |
| **Ethnicity** | Oromo | 404 | 91.4 |
| | Amhara | 16 | 3.6 |
| | Other*** | 22 | 5.0 |
| **Maternal education** | No formal education | 44 | 10.0 |
| | Primary (1–8) | 115 | 26.0 |
| | Secondary (9–12) | 133 | 30.1 |
| | College and above | 150 | 33.9 |
| **Husband's education** | No formal education | 21 | 4.8 |
| | Primary (1–8) | 88 | 19.9 |
| | Secondary (9–12) | 121 | 27.4 |
| | College and above | 212 | 48.0 |
| **Occupation** | Housewife | 209 | 47.3 |
| | Government | 93 | 21.0 |
| | Private(owned/employed) | 79 | 17.9 |
| | Farmer | 35 | 7.9 |
| | Student | 22 | 5.0 |
| | Other**** | 4 | 0.9 |
| **Monthly household income** | <1000 | 34 | 7.7 |
| | 1000–5000 | 247 | 55.9 |
| | >5000 | 161 | 36.4 |

* Single, Divorced **Muslim, Wakefata, Adventist; *** Tigre, Gurage, Hadiya; **** Daily labor.

ultrasound. Most of the women 387(87.6%) knew that prenatal ultrasound helped to assess the well-being of the fetus, and 251(56.8%) women knew that a timely prenatal ultrasound scan helped determine the expected date of delivery (Fig 3).

## Attitude of women toward prenatal ultrasound

The mean score of mothers' attitudes towards prenatal ultrasound was 41.65(SD ± 6.85) with minimum and maximum scores of 15 and 50, respectively. About half 224(50.7%) had a positive attitude toward prenatal ultrasound. The majority of 406(91.9%) women reported that ultrasound examinations during pregnancy were crucial. However, 88(19.9%) women believed that prenatal ultrasound scans could potentially result in congenital abnormalities (Table 4).

**Table 2. Timing of first prenatal ultrasound and related factors among women who gave birth in Ambo Town Public Health Institution, Ethiopia, 2022. (N = 442).**

| Variables | Category | Frequency | Percentage (%) |
|---|---|---|---|
| Timing of first prenatal ultrasound | Early | 334 | 71 |
| | Late | 108 | 29 |
| Heard appropriate time of prenatal ultrasound | Yes | 227 | 51.4 |
| | No | 215 | 48.6 |
| Source of information | Health professional | 167 | 37.8 |
| | Family | 27 | 6.1 |
| | Other$ | 33 | 7.5 |
| Gestational age of 1st prenatal ultrasound | 1st trimester | 32 | 7.2 |
| | 2nd trimester | 365 | 82.6 |
| | 3rd trimester | 45 | 10.2 |
| Who requested | Herself | 59 | 13.3 |
| | Health professional | 381 | 86.2 |
| | Other$$ | 2 | 0.5 |
| Number of scans | 1-2 times | 267 | 60.4 |
| | >3 times | 175 | 39.6 |
| Had history ultrasound scan | Yes | 181 | 41.0 |
| | No | 82 | 18.6 |
| Estimated waiting time to get prenatal ultrasound | <30 min | 105 | 23.8 |
| | >30 min | 337 | 76.2 |
| Have got explanation about my scan result | Yes | 248 | 56.5 |
| | No | 191 | 43.5 |

$ Media (TV, Radio, Newspaper), Internet; $$ Partner or spouse.

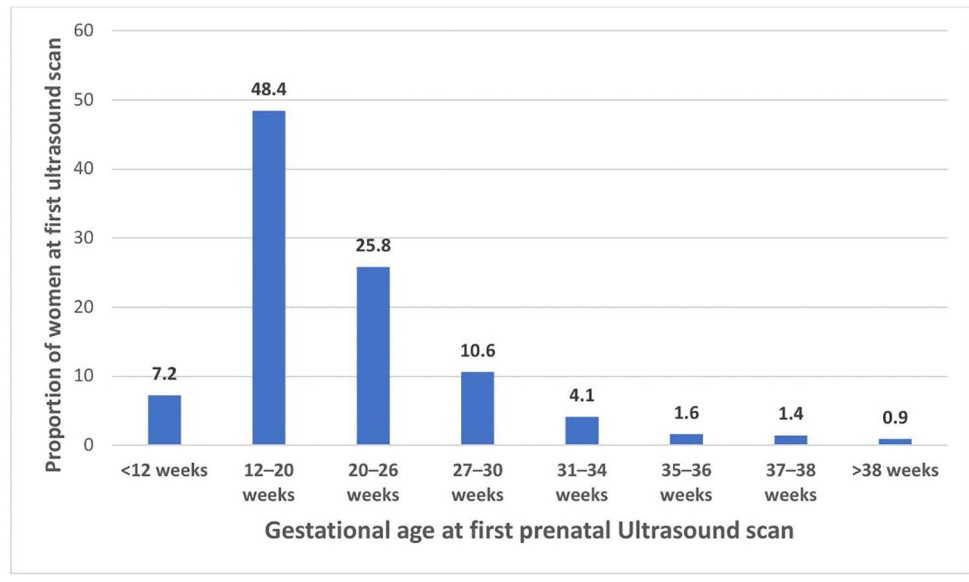

**Fig 2. Timing of the first prenatal Ultrasound scan among Women who gave birth in Public Health institutions, Ambo Town, Ethiopia, 2022 (N = 442).**

**Table 3. Obstetrics and reproductive health related factors among women who gave birth in Ambo Town Public Health Institution, Ethiopia, 2022. (N = 442).**

| Variables | Category | Frequency | Percentage (%) |
|---|---|---|---|
| Intended pregnancy | Yes | 416 | 94.1 |
| | No | 26 | 5.9 |
| Parity | Primipara | 192 | 43.4 |
| | Multipara | 250 | 56.6 |
| Place of ANC follow up | Hospital | 293 | 66.3 |
| | Health center | 134 | 30.3 |
| | Private clinic | 15 | 3.4 |
| Gestational age at 1st ANC follow-up | 1st trimester | 120 | 27.1 |
| | 2nd trimester | 297 | 67.2 |
| | 3rd trimester | 25 | 5.7 |
| Number of times women had ANC contact | 1–2 | 66 | 14.9 |
| | 3–4 | 310 | 70.1 |
| | ≥5 | 66 | 14.9 |
| Number of times husband presented in ANC contact | Nil | 138 | 31.2 |
| | 1–2 | 164 | 37.1 |
| | 3–4 | 117 | 26.5 |
| | ≥5 | 23 | 5.2 |
| Health problem before pregnancy | Yes | 28 | 6.3 |
| | No | 414 | 93.7 |
| Health problem during pregnancy | Yes | 51 | 11.5 |
| | No | 391 | 88.5 |
| Alcohol use in recent pregnancy | Yes | 13 | 2.9 |
| | No | 429 | 97.1 |
| Had bad obstetric history | Yes | 39 | 8.9 |
| | No | 403 | 91.1 |
| Place of delivery | Hospital | 382 | 86.4 |
| | Health center | 60 | 13.6 |
| Mode of delivery | Vaginal | 322 | 72.9 |
| | Caesarean section | 120 | 27.1 |
| Previous history of caesarean section | Yes | 39 | 8.9 |
| | No | 231 | 52.6 |
| Number of babies delivered | Singleton | 436 | 98.6 |
| | Twin | 6 | 1.4 |
| Newborn outcomes | Alive | 437 | 98.9 |
| | Dead | 5 | 1.1 |
| Gestational age at birth | <37 wks | 15 | 3.4 |
| | 37 wks–42 wks | 423 | 95.7 |
| | ≥42 wks | 4 | 0.9 |
| Birth weight | <2500 g | 15 | 3.4 |
| | 2500–4000 g | 404 | 91.4 |
| | ≥4000 g | 23 | 5.2 |
| Admitted to Neonatal intensive care unit | Yes | 46 | 10.4 |
| | No | 396 | 89.6 |

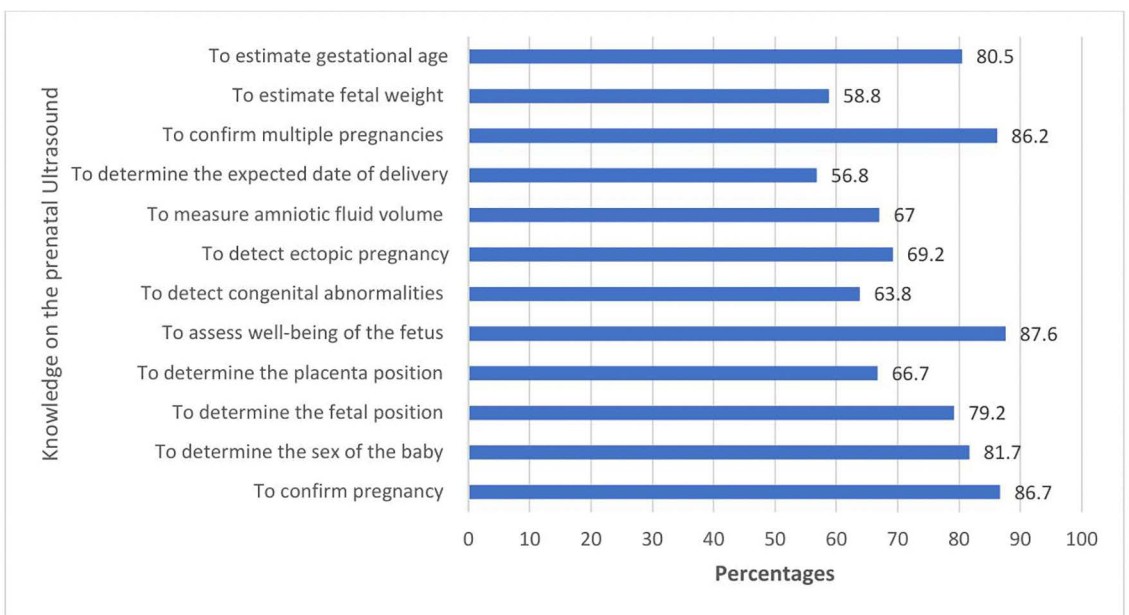

**Fig 3. Knowledge on prenatal ultrasound among women who gave birth in Ambo Town Public Health Institution, Ethiopia, 2022 (N=442).**

**Table 4. Attitude toward prenatal ultrasound among women who gave birth in Ambo Town Public Health Institution, Ethiopia, 2022. (N=442).**

| Variables | Yes | Percent (%) |
|---|---|---|
| Perceived that Prenatal ultrasound is safe for mother | 371 | 83.9 |
| Perceived that Prenatal ultrasound is safe for fetus | 352 | 79.6 |
| Perceived that Knowing the sex of your child before birth will bring bad luck | 23 | 5.2 |
| Believe that Prenatal ultrasound can lead to congenital anomaly. | 88 | 19.9 |
| Perceived to terminate the pregnancy if the sex of the child is other than you prefer | 23 | 5.2 |
| Perceived to Educate others about prenatal ultrasound is important | 301 | 68.1 |
| Believe that Prenatal ultrasound can lead to cancer | 32 | 7.2 |
| Believe that Prenatal ultrasound gives accurate information | 296 | 67.0 |
| Believe that Prenatal ultrasound tends to be offered by chance | 88 | 19.9 |
| Perceived that prenatal ultrasound in an essential investigation during pregnancy | 406 | 91.9 |
| Favorable Attitude | 224 | 50.7 |
| Unfavorable Attitude | 218 | 49.3 |

## Factors associated with timing of first prenatal ultrasound

In bivariate analysis, residence, maternal education, age, monthly household income, gravidity, intended pregnancy, place of ANC follow-up, previous prenatal ultrasound history, poor obstetric history, maternal frequency of ANC contact, level of knowledge of prenatal ultrasound, and level of attitude toward prenatal ultrasound were found to be associated with having a timely prenatal ultrasound scan (P-value<0.25). However, in the multivariate analysis, residence, previous history of prenatal ultrasound, place of ANC follow-up, and knowledge of prenatal ultrasound were significantly associated with the timing of the first prenatal ultrasound (P-value<0.05).

The likelihood of having a timely prenatal ultrasound scan among women residing in urban areas was nearly six times higher than that among women residing in rural areas (AOR = 5.64, 95%CI = 2.53–12.55). Respondents with a history of prenatal ultrasound in previous pregnancies were twice as likely to undergo prenatal ultrasound in a timely manner (AOR = 2.47, 95%CI:1.24–4.89). Similarly, respondents who attended ANC visits at hospitals were four times more likely to have early prenatal ultrasound scans than their counterparts (AOR = 4.46, 95%CI: 2.26–8.81). Those postpartum women who had good knowledge of ultrasound during pregnancy were twice as likely to have a timely prenatal ultrasound scan than those with poor knowledge (AOR = 2.13, 95% CI: 1.02–4.41) ([Table 5]).

**Table 5. Bivariate and multivariate analysis on timing of first prenatal ultrasound and associated factors among women who gave birth in Ambo town public health institutions, Ethiopia, 2022.**

| Variable | Category | Timing of first prenatal ultrasound | | COR at CI 95% | AOR at CI 95% | P-value |
|---|---|---|---|---|---|---|
| | | Early | Late | | | |
| Residence | Urban | 289 | 63 | 11.92 (6.98–20.38) | 5.64 (2.53–12.55) | 0.000* |
| | Rural | 25 | 65 | 1 | 1 | |
| Maternal education | No formal education | 19 | 25 | 1 | 1 | |
| | Primary (1 –8) | 64 | 51 | 1.65 (0.81–3.32) | 0.82 (0.26–2.60) | 0.749 |
| | Secondary (9 –12) | 98 | 35 | 3.68 (1.81–7.49) | 1.02 (0.29–3.60) | 0.970 |
| | College & above | 133 | 17 | 10.29 (4.71–22.48) | 1.65 (0.43–6.37) | 0.463 |
| Age | <20 | 21 | 6 | 1 | 1 | |
| | 20–25 | 120 | 47 | 4.13 (1.23–13.89) | 0.83 (0.17–3.98) | 0.746 |
| | 26–29 | 112 | 42 | 3.01 (1.26–7.20) | 0.24 (0.24–3.57) | 0.928 |
| | 30–34 | 50 | 20 | 3.15 (1.31–7.58) | 0.55 (0.15–2.00) | 0.370 |
| | ≥35 | 11 | 13 | 2.95 (1.13–7.68) | 0.76 (0.19–2.94) | 0.697 |
| Household monthly income | <1000 | 10 | 24 | 1 | 1 | |
| | 1000–5000 | 166 | 81 | 4.91 (2.24–10.77) | 2.28 (0.54–9.53) | 0.258 |
| | >5000 | 138 | 23 | 14.40 (6.09–34.01) | 3.66 (0.19–17.10) | 0.099 |
| Gravidity | Primigravida | 133 | 47 | 1 | 1 | |
| | Multigravida | 181 | 81 | 1.26 (0.82–1.93) | 1.16 (0.65–2.07) | 0.610 |
| Intended pregnancy | Yes | 303 | 113 | 0.27 (0.12–0.61) | 2.79 (0.81–9.59) | 0.103 |
| | No | 11 | 15 | 1 | 1 | |
| Place of ANC follow up | Hospital | 242 | 51 | 5.85 (3.71–9.22) | 4.46 (2.26–8.81) | 0.000* |
| | Private clinic | 12 | 3 | 4.93 (1.33–18.28) | 1.60 (0.14–17.1) | 0.698 |
| | Health center | 60 | 74 | 1 | 1 | |
| *Previous history of ultrasound* | Yes | 142 | 39 | 0.24 (0.14–0.43) | 0.51 (0.12–2.18) | 0.371 |
| | No | 39 | 43 | 1 | 1 | |
| Bad obstetrics history | Yes | 24 | 17 | 1.85 (0.95–3.57) | 0.68 (0.27–1.72) | 0.419 |
| | No | 290 | 111 | 1 | 1 | |
| Maternal ANC frequency | 1–2 | 40 | 26 | 1 | 1 | |
| | 3–4 | 216 | 94 | 1.49 (0.86–2.58) | 0.78 (0.29–2.07) | 0.620 |
| | ≥5 | 58 | 8 | 4.71 (1.93–11.46) | 1.47 (0.36–5.96) | 0.586 |
| Level of knowledge | Poor knowledge | 79 | 74 | 1 | 1 | |
| | Good knowledge | 235 | 54 | 4.07 (2.64–6.28) | 2.13 (1.02–4.41) | 0.042* |
| Level of attitude | Negative attitude | 138 | 80 | 1 | 1 | |
| | Positive attitude | 176 | 48 | 2.12 (1.39–3.24) | 0.84 (0.41–1.74) | 0.651 |

*p-value<0.05, CI: confidence interval, AOR: Adjusted odd ratio, COR: Crude odd ratio.

## Discussion

This study assessed the timing of the first ultrasound scanning and its factors in women who delivered at health facilities in Ambo town, Ethiopia. The proportion of women who utilized prenatal ultrasound in a timely manner was 71.0% [(95% CI); (67.0%−75.6%)]. These findings align with the study conducted in Jimma Town, Ethiopia, which reported a prenatal ultrasound utilization rate of 76.1% [17]. A possible explanation could be that both studies took place in the town with little difference in the study period, and most participants (80%) were urban residents. In addition, public health institutions have integrated working systems, as women who follow their ANC at health centers are linked to hospitals for ultrasound services.

Conversely, the results of this study was higher than the study conducted at 25 health centers in four region of Ethiopia (27.4%) [26]. This may be due to differences in health institution infrastructure, as this study included referral and general hospitals, in addition to health centers. Similarly, the current finding is also higher than that of a study conducted in Kenya (37.9%) [26]. This may be because the Kenya study was conducted in rural and peri-urban areas, and the time variation between the studies.

Compared to other studies, the current finding was lower than studies conducted in Canada (95.8%) [24], Iceland (95%) [27], and China (96.1%) [28]. This difference might be due to socio-demographic and socio-economic variations between the countries. This discrepancy may be attributed to variations in the healthcare service system.

Regarding associated factors, the current study found a strong association between the timing of the first prenatal ultrasound and residency. Women from urban residences were six times more likely to undergo timely prenatal ultrasound than those from rural areas. This finding was similar to that of a study conducted in Jimma Town [17]. This might be because those from urban areas had better access to information and knowledge regarding the optimal timing for initial prenatal ultrasound, as well as a higher availability of ultrasound machines compared to rural areas [15].

Maternal knowledge of prenatal ultrasound is another explanatory variable that is associated with the timing of the first ultrasound scan. Mothers with good overall knowledge of prenatal ultrasound were twice as more likely to use initial prenatal ultrasound in a timely manner compared to their counterparts. This finding aligns with those of studies conducted in Ethiopia [15,25], Uganda [16] and Nigeria [20]. This could be because individuals with a strong understanding of prenatal ultrasound are aware of its benefits and proactively seek it out, even without a doctor's recommendation.

The history of prenatal ultrasound scanning was found to significantly enhance timely prenatal ultrasound use. Respondents who had a history of prenatal ultrasound in a previous pregnancy were twice as likely to use the first prenatal ultrasound in a timely manner than their counterparts. A consistent finding was reported in studies conducted in Canada [24] and Kenya [18]. This may be explained by the fact that multigravida mothers have more experience with prenatal ultrasound because they are familiar with the service. Moreover, based on their experience, they might have a greater desire or expectation for a healthy birth outcome, making them more likely to seek prenatal ultrasound in a timely manner.

The study also revealed that the location of the ANC follow-up was significantly associated with the timing of the ultrasound scan. Women who attended ANC at the hospital were four times more likely to undergo the first prenatal ultrasound in a timely manner than those who had attended other health facilities. This is easily explained by the fact that the ultrasound machine is only available in the hospital; therefore, those who attend ANC at the hospital could easily and frequently receive the service. A qualitative study also supports this finding, as the unavailability of skilled providers at health centers was a barrier to timely prenatal ultrasound scans [29].

The results revealed that attitude was not found significantly associated with the early use of ultrasound among women in the multivariable model. This could be attributed to the fact that nearly half of the participants held an unfavorable attitude towards prenatal ultrasound, and some rumors about its potential effects on the fetus continued to circulate among the women. Additionally, structural and health system barriers such as the late initiation of antenatal care and the limited availability of ultrasound services may impede early uptake regardless of attitude. Consequently, this finding highlights the

necessity of complementing attitude-focused messaging with targeted interventions that address these practical barriers and emphasizes the importance of early antenatal care attendance and having early prenatal ultrasound [15,30,31].

These findings imply that, the health centers, obstetric care providers and policymakers at different level should prioritize and work on making prenatal ultrasound services accessible to all women at the health centers or through early referral to where the service is available. Moreover, awareness creation activities must also be considered for improving timely prenatal ultrasound scan.

This study had the following main limitations. The study used a cross-sectional design, which makes it difficult to establish causality. Furthermore, as the study exclusively included women who underwent ultrasound scans and delivered at public health facilities in Ambo Town, the results may not be generalizable to women who gave birth at home or did not receive an ultrasound scan. Recall bias might be introduced for some variables that require women to remember past events to respond. The current study does not consider factors related to health facilities and care providers, which could be important to comprehensively explore the factors that could affect perinatal ultrasound scans and their timing.

## Conclusion

The study found that less than three-fourths of women received timely prenatal ultrasound. The factors associated with timely ultrasound scanning were urban residence, history of prenatal ultrasound, attending ANC at the hospital, and good knowledge of prenatal ultrasound. Therefore, policymakers and other stakeholders should prioritize the availability of prenatal ultrasound services at the health center or through early referral to the hospital, creating awareness on early prenatal ultrasound as a critical intervention for reducing maternal and perinatal morbidity and mortality. Furthermore, conducting large-scale research to identify the relationship between the frequency of ultrasound scans and its outcomes and addressing health facilities and care provider-related factors are recommended.

## Supporting information

**S1 File. English version questionnaire.**
(PDF)

**S2 File. SPSS data file.**
(SAV)

## Acknowledgments

We are thankful to the Ambo Town Health office, and the staff of health facilities in Ambo Town for their demonstrative support during data collection. We are also grateful to the study participants for their kind cooperation.

## Author contributions

**Conceptualization:** Samson Mesfin, Gizachew Abdissa Bulto, Motuma Gutu.

**Data curation:** Samson Mesfin, Gizachew Abdissa Bulto, Motuma Gutu.

**Formal analysis:** Samson Mesfin, Gizachew Abdissa Bulto, Motuma Gutu, Natnael Dechasa Gemeda.

**Funding acquisition:** Samson Mesfin.

**Investigation:** Samson Mesfin, Gizachew Abdissa Bulto, Natnael Dechasa Gemeda.

**Methodology:** Samson Mesfin, Gizachew Abdissa Bulto, Motuma Gutu, Natnael Dechasa Gemeda.

**Project administration:** Samson Mesfin, Gizachew Abdissa Bulto, Natnael Dechasa Gemeda.

**Resources:** Samson Mesfin.

**Software:** Samson Mesfin, Gizachew Abdissa Bulto, Motuma Gutu, Natnael Dechasa Gemeda.

**Supervision:** Samson Mesfin, Gizachew Abdissa Bulto, Motuma Gutu, Natnael Dechasa Gemeda.

**Validation:** Samson Mesfin, Gizachew Abdissa Bulto, Motuma Gutu.

**Visualization:** Samson Mesfin, Gizachew Abdissa Bulto, Motuma Gutu, Natnael Dechasa Gemeda.

**Writing – original draft:** Samson Mesfin, Gizachew Abdissa Bulto, Motuma Gutu, Natnael Dechasa Gemeda.

**Writing – review & editing:** Samson Mesfin, Gizachew Abdissa Bulto, Motuma Gutu, Natnael Dechasa Gemeda.

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
