## [Decision Letter · Decision Letter 0]

1 Aug 2025

Dear Dr. Bulto,

Thank you for submitting your manuscript to PLOS ONE. After careful consideration, we feel that it has merit but does not fully meet PLOS ONE’s publication criteria as it currently stands. Therefore, we invite you to submit a revised version of the manuscript that addresses the points raised during the review process.

We look forward to receiving your revised manuscript.

Kind regards,

Ahmed Mohamed Maged, MD

Academic Editor

PLOS ONE

Journal Requirements:

2. Please note that funding information should not appear in the Acknowledgments section or other areas of your manuscript. We will only publish funding information present in the Funding Statement section of the online submission form. Please remove any funding-related text from the manuscript. 

3. In the online submission form, you indicated that:

“The datasets used and/or analyzed during the current study are available from the corresponding author on reasonable request.”

**Additional Editor Comments:**

Please respond to all reviewers comments.

Reviewers' comments:

Reviewer's Responses to Questions

**Comments to the Author**

1. Is the manuscript technically sound, and do the data support the conclusions?

Reviewer #1: Yes

Reviewer #2: Yes

Reviewer #3: Yes

Reviewer #4: Yes

Reviewer #5: Yes

Reviewer #6: Partly

2. Has the statistical analysis been performed appropriately and rigorously?

Reviewer #1: Yes

Reviewer #2: Yes

Reviewer #3: Yes

Reviewer #4: Yes

Reviewer #5: Yes

Reviewer #6: Yes

3. Have the authors made all data underlying the findings in their manuscript fully available?

Reviewer #1: Yes

Reviewer #2: Yes

Reviewer #3: Yes

Reviewer #4: Yes

Reviewer #5: Yes

Reviewer #6: No

4. Is the manuscript presented in an intelligible fashion and written in standard English?

Reviewer #1: Yes

Reviewer #2: Yes

Reviewer #3: Yes

Reviewer #4: Yes

Reviewer #5: Yes

Reviewer #6: Yes

Reviewer #1: The paper is well-written and offers valuable insights into the existing body of knowledge; however, I have some concerns that require attention.

1. In the abstract's background section, it is better to change the statement “Hence, it was aimed to assess…” with “Hence, the study aimed to assess…”

2. Similarly in the abstract's result section write the variable as “living in urban area”

3. Again, in the abstract's conclusion section, change the term “place of residence” to “urban residence” to make them consistent.

4. In the background, you repeated this statement, “Prenatal ultrasound is an essential core package of routine Antenatal care (ANC)“, two times.

5. Again in the background why you use the term furthermore in “Furthermore, the introduction…?

6. In the method part, it is better to change the sentence “All women who delivered at public health facilities in Ambo town and randomly selected were the study population.” with "All randomly selected women who delivered at public health facilities in Ambo town were the study population."

7. You use the mean and standard deviation as summary measures in all documents. Did you check your data distribution? You used the mean value as a cut-off point to categorise the knowledge and attitude. Did you check your data distribution for these variables?

8. Add references for the information you use in the study area and merge some unmerged references like in the background, method… Check the whole document.

9. How did you handle cases where a woman received antenatal care (ANC) at another health facility and delivered at one of the study facilities during your study period?

10. Is assessing women’s knowledge and attitude toward prenatal ultrasound at the time of delivery not influenced by the kind of care they received throughout their pregnancy, as all women had a history of ANC? Could you please explain why the study was not conducted while they were receiving ANC?

Reviewer #2: This study employed a health-facility-based cross-sectional design to assess the timing of first prenatal ultrasound and associated factors in Ambo Town, central Ethiopia . While the methodology has several strengths, some aspects needs critical consideration.

Rationale:

The introduction did not contextualize why Ambo Town was selected for this study (e.g., urban-rural disparities in healthcare access) and clarify if prior local data existed on this topic.

Study Design and Setting

This study covers a relatively short period (September 12 to October 30, 2022) and captures a snapshot, however it might not account for seasonal variations or other temporal factors that could influence ultrasound utilization. (could be addressed in limitation)

Tool Development:

Since the questionnaire were self-developed, how was the construct validity ensured? How was it considered enough to assess the knowledge and attitude? This study talks about language validity and internal consistency for intended constructs but not mentioned regarding how many items would make eligible construct for the knowledge and attitude. (explain the method)

BIAS and confounders addressing:

Exclusion Criteria: Women who had no prenatal ultrasound scan at all during their recent pregnancy were excluded. While this focuses the study on the timing of first ultrasound, it means the study does not capture the portion of the population that does not receive any ultrasound, which could be a significant public health concern. Selection bias.

Unadjusted variables like healthcare provider density or facility resources (ultrasound availability) could influence results. How was this addressed? (potential confounders)

Facility-Based Study Design: The findings may not be applicable to women who delivered at home or in private health facilities creating selection bias. Is this a limitation? Because generally, the percentage of women delivering in private facility might be much higher than home delivery. Clarify the local delivery practices to give the reader a clear picture.

Recall Bias in Reporting of Ultrasound Timing

The study tried to minimize the recall bias by cross checking with record card, however, the study did not explicitly report on discrepancies found between interview responses and medical records and how it was resolved regarding ultrasound timing.

Discussion:

The study overstates causality while explaining implications, such as "good knowledge" leads to timely scans” in absence of longitudinal data.

The study does not explore why health centers underperform for e.g., lack of machines, trained staff etc.

Review

This study employed a health-facility-based cross-sectional design to assess the timing of first prenatal ultrasound and associated factors among women who delivered in Ambo Town, central Ethiopia. While the methodology demonstrates several strengths, such as systematic random sampling, robust sample size calculation, and use of validated tools, certain aspects needs some consideration to enhance rigor and generalizability.

Rationale: The introduction effectively establishes the importance of timely prenatal ultrasound per WHO and Ethiopian guidelines. However, it does not explicitly justify the selection of Ambo Town, particularly regarding urban-rural disparities in healthcare access or whether prior local studies on this topic exist. Clarifying these points would strengthen the study’s relevance to regional policy and practice.

Study Design: The data collection period (September 12 to October 30, 2022) provides a snapshot of ultrasound utilization but may not account for seasonal variations (e.g., agricultural cycles affecting healthcare access) or other temporal factors. This is not acknowledged in the limitation.

Tool Development and Validity

The study used a self-developed questionnaire, with translation and back-translation ensuring language validity and Cronbach’s alpha (0.85–0.87) confirming internal consistency. However, the manuscript does not detail how construct validity was established (e.g., expert review, factor analysis) or justify the sufficiency of the 12-item knowledge scale and 10-item attitude scale. A clearer description of item selection and validation methods would add up to the validity and reliability of the tool.

Selection Bias:

Exclusion of Women Without Ultrasound: The study excluded women who had no prenatal ultrasound, omitting a critical population segment that may face systemic barriers (e.g., poverty, distance). This introduces selection bias and limits understanding of non-utilization drivers.

Facility-Based Design: Findings may not generalize to women delivering at home or in private facilities. The manuscript should clarify local delivery practices (e.g., percentage of home births) to contextualize this limitation.

Unmeasured Confounders:Facility-level factors (e.g., ultrasound machine availability, staff training) were not adjusted for, potentially confounding associations (e.g., hospital ANC attendance vs. health centers). Acknowledging these unmeasured confounders in limitation would shed light on scope of the utility of the findings.

Recall Bias: While the study cross-checked self-reported ultrasound timing with medical records, it did not report discrepancies or resolution methods. Quantifying discordance rates would strengthen validity claims.

Discussion: The discussion appropriately compares findings with global and regional studies but overstates causal relationships (e.g., "good knowledge leads to timely scans") without longitudinal or qualitative data. Additionally, it does not explore structural barriers (e.g., health center resource gaps) that may explain underperformance. Addressing these gaps would provide actionable insights for policymakers.

Reviewer #3: I had gone through “Timing of first prenatal Ultrasound and associated factors among women who gave birth at health institutions in Ambo Town, central Ethiopia.” Few minor modifications need to be done

1. Title can be modified as ‘Timing of first prenatal Ultrasound and factors associated with it among women who gave birth at health institutions in Ambo Town, central Ethiopia.’

2. In result part the author had mentioned about “The line graph below shows the proportion of mothers who had their first prenatal ultrasound scan along with the new 8 contact ANC model (Fig. 1). It would be better if the author give an introduction to new 8 contact ANC model in the background, so that reviewers or readers understand it well. The title of Fig 1 is confusing can be modified. The figure shows the proportion of women taking first ultrasound in various weeks of gestation.

3. In knowledge part one of the question is to determine the sex of the baby. Usually in India sex determination through prenatal USG is prohibited or punishable act. I don’t know about the rules and regulation of Ethiopia Government.

Reviewer #4: This manuscript addresses an important aspect of maternal health services—timely utilization of prenatal ultrasound and identifies associated factors within the Ethiopian context. The study is relevant and aligns with both national and global efforts to reduce maternal and neonatal morbidity and mortality. However, several points require improvement and refinement as follows:

1. Novelty and Scientific Contribution

The authors should clarify the novelty of this study compared to previous studies conducted in other regions of Ethiopia (such as Jimma) or other countries. What makes this study distinct, and how do the findings contribute to strengthening local or regional health policies? Additionally, the authors are encouraged to justify why Ambo Town was chosen as the study location. Is it due to suboptimal ANC services or high maternal mortality in the area?

2. Inconsistency Between p-value and Confidence Interval (CI)

In the multivariate analysis, the variable “previous history of prenatal ultrasound” shows an Adjusted Odds Ratio (AOR) of 1.20 with a p-value of 0.009, but the confidence interval (CI: 0.69–2.08) includes the value 1. This is statistically inconsistent because an effect should not be considered significant if the CI includes 1. The authors are requested to verify and correct the data or provide a clear explanation for this discrepancy.

3. Attitude as a Non-significant Factor

Although the assessment of attitudes toward prenatal ultrasound is presented, the results indicate that attitude was not significantly associated with the timely use of ultrasound. The authors should discuss why this may be the case and consider the implications for community health education strategies.

4. Study Limitations

Although limitations are briefly mentioned, they should be expanded, particularly regarding potential selection bias (since the study only included women who gave birth in public health facilities) and possible recall bias, even though medical records were also used.

Reviewer #5: Thank you for writing this interesting paper on an important issue. I enjoyed reading it. One of the most important benefits of ultrasound regarding ascertainment of the expected date of delivery is the avoidance of pregnancies that proceed past 41 weeks, as these prolonged pregnancies carry risks of stillbirth and other complications such as shoulder dystocia associated with large birthweight. I feel that you should mention this issue of avoidance of prolonged pregnancy by ascertainment of dates in pregnancies where women are not aware of their dates. Also perhaps you could enhance the discussion by considering more about the reasons why dates may be so uncertain, for example, breastfeeding a previous baby, certain contraceptives etc.

It would also be pertinent to discuss the financial aspects of US. Who pays for scans? Is it out of pocket? If so this may be a major reason for not being able to attend for a scan.

What happens when anomalies are detected on US? Is there an option to undergo safe abortion if significant anomalies are detected? This is another issue that may affect a woman's choice to attend for US.

In the introduction you talk about problems being "tackled". This is a rather strange word to use in this context and maybe you could find an alternative?

Otherwise there are some minor grammatical errors but these do not affect the readability or comprehension of the paper which is f=good overall.

Reviewer #6: This paper addresses a significant public health issue concerning Ethiopian women, highlighting the importance of prenatal ultrasound as a tool for both prenatal diagnosis and safe motherhood. However, several critical limitations must be addressed.

The most substantial concern is the absence of a coherent conceptual framework. In the Introduction, the author emphasizes the role of early antenatal ultrasound in improving perinatal outcomes and reducing maternal mortality. Yet, there is no discussion of the structural barriers limiting access to this service in the region—arguably the central focus of the study.

Methodologically, the study adopts a cross-sectional design across multiple centers in the Ambo region but inexplicably excludes women who did not undergo antenatal ultrasound. More problematically, the sample size calculation employs a double proportion population formula, using a history of previous abortion as the "risk factor" to compare exposed and unexposed cases. Only later do the authors introduce "knowledge of prenatal ultrasound" as a variable worth examining among other potential factors influencing the timing of the first prenatal scan.

In essence, there is a disconnect between the introduction, sample size justification, and variable selection. This inconsistency has a cascading effect on the results, where the timing of the first prenatal ultrasound abruptly becomes the outcome variable—apparently the true research question—while a multivariate analysis is conducted on demographic and socioeconomic variables that were neither theoretically grounded nor operationalized (with the exception of prior ultrasound knowledge).

Given these fundamental methodological flaws, a detailed discussion of the results is unwarranted. However, several stylistic issues should be corrected:

Numerical presentation: If numbers and percentages are already provided in parentheses, spelling out the number is redundant (e.g., "262 (59.3%)" suffices; "two hundred sixty-two (59.3%)" is unnecessary).

Informal phrasing: Colloquial expressions (e.g., "almost a half") should be replaced with precise academic language.

Conclusion: The current conclusion is overly lengthy, reiterates results unnecessarily, and obscures the key takeaway. A more concise and impactful statement would be: "Policymakers and stakeholders should prioritize raising awareness about early prenatal ultrasound as a critical intervention for reducing maternal and perinatal morbidity and mortality."

Figure 1: A column chart would more effectively display proportions across gestational age categories than the current continuous line, which is better suited for time-series data.

Finally, the manuscript requires thorough stylistic revision, ideally by a native English speaker, as recurrent grammatical errors detract from its overall clarity and professionalism.

**Do you want your identity to be public for this peer review?** For information about this choice, including consent withdrawal, please see our Privacy Policy

Reviewer #1: No

Reviewer #2: **Yes: ** Dr Sailesh Bhattarai

Reviewer #3: No

Reviewer #4: No

Reviewer #5: No

Reviewer #6: **Yes: ** Juan Carlos Bello-Muñoz

---

## [Author Response · Author response to Decision Letter 1]

17 Aug 2025

PONE-D-25-08139

Timing of first prenatal Ultrasound and associated factors among women who gave birth at health institutions in Ambo Town, central Ethiopia.

PLOS ONE

To: Editor of PLOS ONE Journal

Dear Sir/Madam,

We wish to remind you of our submission of the manuscript entitled “Timing of First Prenatal Ultrasound and Factors Associated with It Among Women Who Gave Birth at Health Institutions in Ambo Town, Central Ethiopia” for publication consideration in PLOS ONE. We express our gratitude for your efforts in reviewing our article. In accordance with the feedback provided by the reviewers and editor, we have revised and modified the manuscript, addressing each comment individually as detailed below. The revised version, with tracked changes and clean version are attached for your reference. The responses to the reviewers' and editor's comments are outlined below.

Gizachew Abdissa Bulto

Email: gizachab@yahoo.com

Corresponding Author  

Editor Comments:

Thank you for submitting your manuscript to PLOS ONE. After careful consideration, we feel that it has merit but does not fully meet PLOS ONE’s publication criteria as it currently stands. Therefore, we invite you to submit a revised version of the manuscript that addresses the points raised during the review process.

Author’s response: We are grateful for your valuable comments, suggestions to enrich our paper. Thank you again for considering our article for revision.

Author’s response: Thank you! We did it accordingly.

Author’s response: Thank you. No change has been made.

Academic Editor

Author’s response: Thank you!

Journal Requirements:

2. Please note that funding information should not appear in the Acknowledgments section or other areas of your manuscript. We will only publish funding information present in the Funding Statement section of the online submission form. Please remove any funding-related text from the manuscript.

3. In the online submission form, you indicated that:

“The datasets used and/or analyzed during the current study are available in the manuscript and details from the corresponding author on reasonable request.”

Authors’ response: Thank you! We have revised the data availability statement in to: “All relevant data are within the paper and its Supporting Information files.”

Author’s response: Thank you, we have included in the revised version.

Author’s response: Thank you, we appreciated!

Reviewers' comments:

Reviewer's Responses to Questions

Comments to the Author

1. Is the manuscript technically sound, and do the data support the conclusions?

Reviewer #1: Yes

Reviewer #2: Yes

Reviewer #3: Yes

Reviewer #4: Yes

Reviewer #5: Yes

Reviewer #6: Partly

2. Has the statistical analysis been performed appropriately and rigorously?

Reviewer #1: Yes

Reviewer #2: Yes

Reviewer #3: Yes

Reviewer #4: Yes

Reviewer #5: Yes

Reviewer #6: Yes

Author’s response: Thank you, we appreciated it!

3. Have the authors made all data underlying the findings in their manuscript fully available?

Reviewer #1: Yes

Reviewer #2: Yes

Reviewer #3: Yes

Reviewer #4: Yes

Reviewer #5: Yes

Reviewer #6: No

Authors response: Thank you, we have included on the revised paper.

4. Is the manuscript presented in an intelligible fashion and written in standard English?

Reviewer #1: Yes

Reviewer #2: Yes

Reviewer #3: Yes

Reviewer #4: Yes

Reviewer #5: Yes

Reviewer #6: Yes

Author’s response: Thank you!

5. Review Comments to the Author

Reviewer #1:

The paper is well-written and offers valuable insights into the existing body of knowledge; however, I have some concerns that require attention.

1. In the abstract's background section, it is better to change the statement “Hence, it was aimed to assess…” with “Hence, the study aimed to assess…”

Authors response: Thank you we have revised it accordingly.

2. Similarly in the abstract's result section write the variable as “living in urban area”

Authors response: We accepted your comment and incorporated in the revised version.

3. Again, in the abstract's conclusion section, change the term “place of residence” to “urban residence” to make them consistent.

Authors response: We accepted your comment and incorporated in the revised version.

4. In the background, you repeated this statement, “Prenatal ultrasound is an essential core package of routine Antenatal care (ANC)“, two times.

Authors response: It was a nice observation. Thank you we have corrected it in the revised version removing the later statement.

5. Again in the background why you use the term furthermore in “Furthermore, the introduction…?

Authors response: Thank you we have omitted the word from the third paragraph of the introduction. Now revised in to: The introduction of a timely prenatal ultrasound service at public hospitals in Ethiopia increases …..

6. In the method part, it is better to change the sentence “All women who delivered at public health facilities in Ambo town and randomly selected were the study population.” with "All randomly selected women who delivered at public health facilities in Ambo town were the study population."

Authors response: Thank you for your suggestion, we accepted your comment and revised accordingly.

7. You use the mean and standard deviation as summary measures in all documents. Did you check your data distribution? You used the mean value as a cut-off point to categorise the knowledge and attitude. Did you check your data distribution for these variables?

Thank Authors response: you for this valid concern. We have checked the data for normality distribution and it was found a normally distributed. We have included the statement which support this point under the ‘Data processing and analysis’ section as: “The normality of the distribution concerning participants' knowledge and attitudes towards perinatal ultrasound was assessed through visual inspection of histograms and Q–Q plots. The findings demonstrated that the data were normally distributed.”

8. Add references for the information you use in the study area and merge some unmerged references like in the background, method… Check the whole document.

Authors response: Thank you very much for your suggestions we revised it through merging the references under the introduction, methods and discussion sections accordingly. Additionally, we added one reference for the information under the study area.

9. How did you handle cases where a woman received antenatal care (ANC) at another health facility and delivered at one of the study facilities during your study period?

Authors response: Thank you for raising this valid concern! If the women clearly recall or have documented evidence of their timing of first ultrasound scan they were included, Otherwise, excluded. We added statement regarding this as: “Women who did not undergo a prenatal ultrasound scan or who could not recall or had no record of their first ultrasound scan during their recent pregnancy were excluded.”

10. Is assessing women’s knowledge and attitude toward prenatal ultrasound at the time of delivery not influenced by the kind of care they received throughout their pregnancy, as all women had a history of ANC? Could you please explain why the study was not conducted while they were receiving ANC?

Authors’ response:

Thank you for raising this concern. We acknowledge that women’s knowledge and attitudes toward prenatal ultrasound could potentially be influenced by the type of care they received and the level of health facility they attended during pregnancy. In our study, we collected information on the number of ANC contacts, facility-related characteristics, and other relevant variables; however, none of these were found to significantly affect the timing of the first ultrasound.

While it would have been possible to assess timing among women during their pregnancy, our primary objective was to examine the timing and frequency of the first prenatal ultrasound in relation to pregnancy outcomes, including birth outcomes. For this reason, data were collected from women after delivery. Although our ability to compare outcomes across categories was limited by small sample sizes for each categories, we found no meaningful differences in the outcomes assessed.

Reviewer #2:

This study employed a health-facility-based cross-sectional design to assess the timing of first prenatal ultrasound and associated factors in Ambo Town, central Ethiopia. While the methodology has several strengths, some aspects needs critical consideration.

Authors response: Thank you very much we appreciate your valuable feedbacks.

Rationale:

The introduction did not contextualize why Ambo Town was selected for this study (e.g., urban-rural disparities in healthcare access) and clarify if prior local data existed on this topic.

Authors response: Thank you! This study was conducted with limited budget as a fulfillment for Master’s degree and could not cover a wider geographical area. But we believe we had also included women from rural areas (20.4%) who came to give birth at those Public Health facilities in Ambo town which includes General and Teaching Hospitals providing services for more than 3 million populations. We have included one statement as a rationale for conducting this study in the current study area. Which reads as: “Whilst national guidelines recommend early prenatal ultrasound (prior to 24 weeks of gestational age), there is a dearth of information regarding its implementation in Ethiopian public health facilities. Furthermore, no data are available on this topic in the study area to guide local interventions. ...”

Study Design and Setting

This study covers a relatively short period (September 12 to October 30, 2022) and captures a snapshot, however it might not account for seasonal variations or other temporal factors that could influence ultrasound utilization. (could be addressed in limitation)

Authors response: Thank you for raising this concern. Although there might be seasonal variations in the utilization of ANC and timing of ultrasound use among women, in the current study we collected data from women who gave birth at public health facilities and we collected data for about seven weeks which could touch three seasons with additional more than a month in the other season. Hence their ultrasound use and timing issue could be addressed within the 10 months duration. It is true that cross sectional study gives us a snapshot of information. We have already included a limitation as: “Additionally, since it only included women who gave birth at public health facilities in Ambo town, the findings may not be generalizable to women who delivered at home.”

Tool Development:

Since the questionnaire were self-developed, how was the construct validity ensured? How was it considered enough to assess the knowledge and attitude? This study talks about language validity and internal consistency for intended constructs but not mentioned regarding how many items would make eligible construct for the knowledge and attitude. (explain the method)

BIAS and confounders addressing:

Authors response: Thank you. We adapted the questionnaire from the previously conducted related literatures. We have checked the internal reliability of the tool with Cronbach alpha test and we found: the Cronbach's alpha value was calculated at 0.87, and 0.85 for attitude assessment items. We have already indicated the number of items included for determining the knowledge status and for the attitude score under the operational definitions.

---

## [Decision Letter · Decision Letter 1]

8 Sep 2025

Dear Dr. Bulto,

Thank you for submitting your manuscript to PLOS ONE. After careful consideration, we feel that it has merit but does not fully meet PLOS ONE’s publication criteria as it currently stands. Therefore, we invite you to submit a revised version of the manuscript that addresses the points raised during the review process.

**ACADEMIC EDITOR: Please respond to all reviewers comments **

We look forward to receiving your revised manuscript.

Kind regards,

Ahmed Mohamed Maged, MD

Academic Editor

PLOS ONE

Journal Requirements:

Reviewers' comments:

Reviewer's Responses to Questions

**Comments to the Author**

Reviewer #1: All comments have been addressed

Reviewer #2: (No Response)

Reviewer #4: All comments have been addressed

Reviewer #5: (No Response)

2. Is the manuscript technically sound, and do the data support the conclusions?

Reviewer #1: Yes

Reviewer #2: Yes

Reviewer #4: Partly

Reviewer #5: Partly

3. Has the statistical analysis been performed appropriately and rigorously?

Reviewer #1: Yes

Reviewer #2: Yes

Reviewer #4: Yes

Reviewer #5: Yes

4. Have the authors made all data underlying the findings in their manuscript fully available?

Reviewer #1: Yes

Reviewer #2: Yes

Reviewer #4: Yes

Reviewer #5: Yes

5. Is the manuscript presented in an intelligible fashion and written in standard English?

Reviewer #1: Yes

Reviewer #2: Yes

Reviewer #4: Yes

Reviewer #5: Yes

Reviewer #1: (No Response)

Reviewer #2: Most of the comments regarding confounders have been addressed and are appropriately framed within the limitations section of this paper. highlighting a need for future research with more comprehensive data collection.

Regarding the knowledge and attitude questions, what was the nature of options in each items? Yes, no, don't know or likert scale? Please clearly mention it what the total highest score would be in each construct.

Thank you

Reviewer #4: 1) The discussion occasionally overstates causality (e.g., "good knowledge leads to timely scans”). Reframe findings as associations only.

2) Expand on health system factors (availability of machines, trained staff) that were not measured but could explain the stronger effect seen for hospital-based ANC.

3) The authors may consider adding the implications of their findings for midwifery practice, for example, highlighting the role of midwives in providing antenatal education and making timely referrals to ensure that pregnant women receive an ultrasound before 24 weeks.

Reviewer #5: I think most of my comments have been addressd. However I still feel that you should at least indicate the proportion of women who do not have any US at all. Otherwise it is a bit misleading to remark on the proportion of women who have timely US as this is not a proportion of the total of all pregnant women. You could say, "of the women who had an US scan, the proportion who ...etc" But even so it would add greatly if you could comment on the proportion of women who did or did not have any US

**Do you want your identity to be public for this peer review?** For information about this choice, including consent withdrawal, please see our Privacy Policy

Reviewer #1: No

Reviewer #2: **Yes: ** Dr Sailesh Bhattarai

Reviewer #4: No

Reviewer #5: No

---

## [Author Response · Author response to Decision Letter 2]

27 Sep 2025

PONE-D-25-08139R1

Timing of first prenatal Ultrasound and associated factors among women who gave birth at health institutions in Ambo Town, central Ethiopia

PLOS ONE

To: Editor of Plos One Journal

Dear Sir/Madam,

We sincerely express our gratitude for your efforts in reviewing our article and forwarding you valuable comments, and suggestions. We are also excited for considering our article further for publication. In accordance with the feedback provided by the reviewers and editor, we have revised and modified the manuscript, addressing each comment individually as detailed below. The revised version, with tracked changes and clean version are attached for your reference. The responses to the reviewers' and editor's comments are outlined below.

and considering our article for revision

Gizachew Abdissa Bulto

Email: gizachab@yahoo.com

Corresponding Author

Editor Comment:

Thank you for submitting your manuscript to PLOS ONE. After careful consideration, we feel that it has merit but does not fully meet PLOS ONE’s publication criteria as it currently stands. Therefore, we invite you to submit a revised version of the manuscript that addresses the points raised during the review process.

Author’s response: Thank you very much for considering our article for publication. We are grateful for your valuable comments, and suggestions.

Author’s response: Thank you! We did it accordingly. We haven’t made change on our financial statement.

Journal Requirements:

Author’s response: Thank you! There is no such recommendation forwarded from the reviewer.

Reviewers' comments:

Reviewer's Responses to Questions

Comments to the Author

1. If the authors have adequately addressed your comments raised in a previous round of review and you feel that this manuscript is now acceptable for publication, you may indicate that here to bypass the “Comments to the Author” section, enter your conflict of interest statement in the “Confidential to Editor” section, and submit your "Accept" recommendation.

Reviewer #1: All comments have been addressed

Reviewer #2: (No Response)

Reviewer #4: All comments have been addressed

Reviewer #5: (No Response)

Author’s response: Thank you very much.

2. Is the manuscript technically sound, and do the data support the conclusions?

Reviewer #1: Yes

Reviewer #2: Yes

Reviewer #4: Partly

Reviewer #5: Partly

Author’s response: Thank you very much. We have addressed your comments further in the revision.

3. Has the statistical analysis been performed appropriately and rigorously?

Reviewer #1: Yes

Reviewer #2: Yes

Reviewer #4: Yes

Reviewer #5: Yes

Author’s response: Thank you very much.

4. Have the authors made all data underlying the findings in their manuscript fully available?

Reviewer #1: Yes

Reviewer #2: Yes

Reviewer #4: Yes

Reviewer #5: Yes

Author’s response: Thank you very much.

5. Is the manuscript presented in an intelligible fashion and written in standard English?

Reviewer #1: Yes

Reviewer #2: Yes

Reviewer #4: Yes

Reviewer #5: Yes

Author’s response: Thank you!

6. Review Comments to the Author

Reviewer #1: (No Response)

Author’s response: Thank you, we are grateful for your earlier comments.

Reviewer #2: Most of the comments regarding confounders have been addressed and are appropriately framed within the limitations section of this paper. highlighting a need for future research with more comprehensive data collection.

Author’s response: Thank you, your comments were invaluable for us to enrich our article.

Regarding the knowledge and attitude questions, what was the nature of options in each items? Yes, no, don't know or likert scale? Please clearly mention it what the total highest score would be in each construct.

Thank you.

Author’s response: Thank you very much. We have incorporated this in the revised version under the operational definition section. We have already added under the result section on page 10.

The revision now read:

“Knowledge of prenatal ultrasound: Twelve questions with yes, no and I don’t responses were used to assess participants' knowledge of prenatal ultrasound. The average score was used to determine the status with the minimum and maximum scores of 0 to 12.” “The attitude score was determined based on a composite value derived from the responses to a 5-point Likert-type scale ranging from strongly agree to strongly disagree with the score ranging from 0 to maximum of 50.”

Reviewer #4: 1) The discussion occasionally overstates causality (e.g., "good knowledge leads to timely scans”). Reframe findings as associations only.

Author’s response: Thank you. We have already stated in the first statement as maternal knowledge is associated with having timing of ultrasound scan. The second statement explains this as the association was twice among those who had overall good knowledge of prenatal ultrasound as compared to others.

2) Expand on health system factors (availability of machines, trained staff) that were not measured but could explain the stronger effect seen for hospital-based ANC.

Author’s response: Thank you, we share your concern too. We have acknowledged these under the limitation section of this article.

3) The authors may consider adding the implications of their findings for midwifery practice, for example, highlighting the role of midwives in providing antenatal education and making timely referrals to ensure that pregnant women receive an ultrasound before 24 weeks.

Author’s response: Thank you. We have accepted your comment and revised it accordingly. We added a statement which read:

“These findings imply that, the health centers, obstetric care providers and policymakers at different level should prioritize and work on making prenatal ultrasound services accessible to all women at the health centers or through early referral to where the service is available. Moreover, awareness creation activities must also be considered for improving timely prenatal ultrasound scan.”

Reviewer #5: I think most of my comments have been addressd. However I still feel that you should at least indicate the proportion of women who do not have any US at all. Otherwise it is a bit misleading to remark on the proportion of women who have timely US as this is not a proportion of the total of all pregnant women. You could say, "of the women who had an US scan, the proportion who ...etc" But even so it would add greatly if you could comment on the proportion of women who did or did not have any US

Author’s response: Thank you, we understand your concern, there are indeed women who do not had prenatal ultrasound during their current pregnancy. But, the current study was specifically focused on the timing of their first prenatal ultrasound among women who had at least one scan during their index pregnancy. While determining the proportion of women who never had an ultrasound would also be valuable for intervention, for us it would not be possible to analyze the timing for those without a scan along with those who had it. For this reason, we excluded those women with no ultrasound from the current study. The current study also didn’t accurately capture the number of women who had no prenatal ultrasound scan.

Dear Reviewers and editor, we are very much thankful for your constructive comments and suggestions which helped us to significantly improved our paper.

Thank you!

---

## [Decision Letter · Decision Letter 2]

15 Oct 2025

Dear Dr. Bulto,

Thank you for submitting your manuscript to PLOS ONE. After careful consideration, we feel that it has merit but does not fully meet PLOS ONE’s publication criteria as it currently stands. Therefore, we invite you to submit a revised version of the manuscript that addresses the points raised during the review process.

**ACADEMIC EDITOR: Please respond to all reviewers comments**

We look forward to receiving your revised manuscript.

Kind regards,

Ahmed Mohamed Maged, MD

Academic Editor

PLOS ONE

Journal Requirements:

Reviewers' comments:

Reviewer's Responses to Questions

**Comments to the Author**

Reviewer #4: All comments have been addressed

Reviewer #5: (No Response)

2. Is the manuscript technically sound, and do the data support the conclusions?

Reviewer #4: Yes

Reviewer #5: No

3. Has the statistical analysis been performed appropriately and rigorously?

Reviewer #4: Yes

Reviewer #5: Yes

4. Have the authors made all data underlying the findings in their manuscript fully available?

Reviewer #4: Yes

Reviewer #5: Yes

5. Is the manuscript presented in an intelligible fashion and written in standard English?

Reviewer #4: Yes

Reviewer #5: Yes

Reviewer #4: (No Response)

Reviewer #5: Thank you for your response but without addressing my comments the entire paper is quite misleading. You make claims about the proportion of women who have timely scans which is just not correct unless you include all the women who deliver at the facility, not just those who have a scan at some time or another. This remains unclear in your paper and as such I cannot recommend publication. Please include some data concerning the numbers of women who had no scan at all. You can compare this to the proportion of those who had a timely scan as compared to a late scan but you must state the total who could have had a scan but did not. Otherwise the claims in the papar as it is written are simply misleading.

**Do you want your identity to be public for this peer review?** For information about this choice, including consent withdrawal, please see our Privacy Policy

Reviewer #4: No

Reviewer #5: No

---

## [Author Response · Author response to Decision Letter 3]

26 Oct 2025

PONE-D-25-08139R2

Timing of first prenatal Ultrasound and associated factors among women who gave birth at health institutions in Ambo Town, central Ethiopia

PLOS ONE

To: Editor of PLOS One Journal

Dear Sir/Madam,

We sincerely express our gratitude for your efforts in reviewing our article and forwarding valuable comments and suggestions. We are also excited for considering our article further for publication. In accordance with the feedback provided by the reviewer and editor, we have revised the manuscript and addressed the reviewer comment individually below. The revised version, with tracked changes and clean manuscript are attached for your reference.

Gizachew A Bulto

Email: gizachab@yahoo.com

Corresponding Author

Reviewers' comments:

Reviewer's Responses to Questions

Comments to the Author

1. If the authors have adequately addressed your comments raised in a previous round of review and you feel that this manuscript is now acceptable for publication, you may indicate that here to bypass the “Comments to the Author” section, enter your conflict of interest statement in the “Confidential to Editor” section, and submit your "Accept" recommendation.

Reviewer #4: All comments have been addressed

Reviewer #5: (No Response)

Author’s response: Dear reviewers, we thank you very much!

2. Is the manuscript technically sound, and do the data support the conclusions?

Reviewer #4: Yes

Reviewer #5: No

Author’s response: We thank you for your comment we have addressed your concern below in detail.

3. Has the statistical analysis been performed appropriately and rigorously?

Reviewer #4: Yes

Reviewer #5: Yes

Author’s response: thank you very much!

4. Have the authors made all data underlying the findings in their manuscript fully available?

Reviewer #4: Yes

Reviewer #5: Yes

Author’s response: Thank you!

5. Is the manuscript presented in an intelligible fashion and written in standard English?

Reviewer #4: Yes

Reviewer #5: Yes

Author’s response: We thank you, your comments have helped us to improve our article!

6. Review Comments to the Author

Reviewer #4: (No Response)

Reviewer #5: Thank you for your response but without addressing my comments the entire paper is quite misleading. You make claims about the proportion of women who have timely scans which is just not correct unless you include all the women who deliver at the facility, not just those who have a scan at some time or another. This remains unclear in your paper and as such I cannot recommend publication. Please include some data concerning the numbers of women who had no scan at all. You can compare this to the proportion of those who had a timely scan as compared to a late scan but you must state the total who could have had a scan but did not. Otherwise the claims in the papar as it is written are simply misleading.

Author’s response: Dear reviewer, we thank you very much for your thoughtful comment. We fully understand and appreciate the concern regarding the inclusion of women who had no ultrasound scans.

Our study, however, was specifically designed to assess the timing of the first prenatal ultrasound and its associated factors among women who gave birth at health institutions in Ambo Town and had at least one prenatal ultrasound. Since the primary outcome variable was the timing of the first scan, women who did not undergo any ultrasound were not eligible for inclusion, as their data would not contribute to the study’s objective.

To prevent any misunderstanding, we have now clarified this point in the manuscript as follows:

Methods section under population on page 5 we revised the statements as:

“All randomly selected women who delivered at public health facilities in Ambo Town and had at least one prenatal ultrasound scan were the study population. Women who did not undergo a prenatal ultrasound scan, could not recall, or had no record of their first ultrasound scan during their recent pregnancy were excluded.”

Finally, we have added a statement in the Limitations section to acknowledge that our findings do not represent the overall coverage of prenatal ultrasound use among all women, rather the timing of ultrasound scan among those who received at least one scan.

The revision now read as:

“This study had the following main limitations. The study used a cross-sectional design, which makes it difficult to establish causality. Furthermore, as the study exclusively included women who underwent ultrasound scans and delivered at public health facilities in Ambo Town, the results may not be generalizable to women who gave birth at home or did not receive an ultrasound scan. Recall bias might be introduced for some variables that require women to remember past events to respond. The current study does not consider factors related to health facilities and care providers, which could be important to comprehensively explore the factors that could affect perinatal ultrasound scans and their timing.”

We hope this clarification adds transparency and prevents potential misinterpretation of our results and addresses the reviewer’s concern to reflect the scope and purpose of our study.

7. PLOS authors have the option to publish the peer review history of their article (what does this mean?). If published, this will include your full peer review and any attached files.

Do you want your identity to be public for this peer review? For information about this choice, including consent withdrawal, please see our Privacy Policy.

Reviewer #4: No

Reviewer #5: No

We are thankful for your comments and suggestions.

---

## [Editor Report · Decision Letter 3]

23 Nov 2025

Timing of first prenatal Ultrasound and associated factors among women who gave birth at health institutions in Ambo Town, central Ethiopia

PONE-D-25-08139R3

Dear Dr. Bulto,

We’re pleased to inform you that your manuscript has been judged scientifically suitable for publication and will be formally accepted for publication once it meets all outstanding technical requirements.

Kind regards,

Ahmed Mohamed Maged, MD

Academic Editor

PLOS ONE
---

## [Editor Report · Acceptance letter]

PONE-D-25-08139R3

PLOS ONE

Dear Dr. Bulto,

I'm pleased to inform you that your manuscript has been deemed suitable for publication in PLOS ONE. Congratulations! Your manuscript is now being handed over to our production team.

Kind regards,

on behalf of

Professor Ahmed Mohamed Maged

Academic Editor

PLOS ONE